# Unusual Complicated Gastric Ulcers

**DOI:** 10.3390/medicina57121345

**Published:** 2021-12-09

**Authors:** Mircea Nicolae Brătucu, Virgiliu-Mihail Prunoiu, Victor Strâmbu, Eugen Brătucu, Maria-Manuela Răvaş, Laurenţiu Simion, Radu Petre

**Affiliations:** 1“Carol Davila” Hospital Surgery, 010731 Bucharest, Romania; Mircea.bratucu@umfcd.ro (M.N.B.); victor.strambu@umfcd.ro (V.S.); Petru.radu@umfcd.ro (R.P.); 2“Alexandru Trestioreanu” Oncological Institute, “Carol Davila” University of Medicine and Pharmacy, 022328 Bucharest, Romania; bratucueugen75@gmail.ro (E.B.); maria-manuela.ravas@drd.umfcd.ro (M.-M.R.); laurentiu.simion@umfcd.ro (L.S.)

**Keywords:** giant ulcers over 2–3 cm, multiple ulcers, complications: haemorrhage or perforation

## Abstract

We here draw attention to a practical issue: the approach to certain unusual gastric ulcers with haemorrhage- or perforation-induced complications. This category of ulcers, i.e., giant (over 2–3 cm) and multiple ulcers, is rarely encountered. We discuss the circumstances determining the occurrence of such lesions, their diverse aetiology and pathogenesis, their common manifestations, and the severity of their evolution. Some of the lesions are benign (chronic or acute ulcers), whereas others are neoplastic: carcinoma, stromal tumours, and lymphomas. In gastric ulcers, the characteristics of this particular and rare category of lesions strictly places them in the surgical field, requiring primary surgical intention. Conservative treatments are not effective in such cases, and preoperative biopsies are not appropriate for emergency interventions. Whether these unusual ulcers are benign or malign, they need to be surgically removed.

## 1. Introduction

Discussions of gastric-ulcer-related problems have become increasingly rare in the literature. The interest in this disease has been declining because of a remarkable reduction in their incidence. Gastric ulcer diagnosis and treatment protocols have not changed in the last 20 years. Gastric ulcer complications (haemorrhage and perforations) are precisely included in the diagnosis and treatment algorithms: endoscopic or trans-arterial catheter embolization–TAE haemostasis. These are currently the standard techniques for treating bleeding ulcers. The therapy approach is as strict for ulcer-related perforations: simple suture or simple suture in association with epiploplasty.

In this article, we present surgical techniques that are different from the standard complicated gastric ulcer treatment. Not all haemorrhagic or perforating gastric ulcers can be solved by the usual treatments according to the protocol’s schemes.

Our article addresses exceptional circumstances: ulcers over 2–3 cm or multiple gastric ulcers with complications. In this case, we have two situations that must be managed: either gastric ulcers with a complications-ridden incipient phase or gastric ulcers that have become “untreatable” due to their evolution to ulcers over 2–3 cm or multiple macroscopic ulcers.

The “no acid, no ulcer” aphorism is accepted in the ulcer pathology field. Physiopathologists and the gastroenterologists agree that normal hydrochloric/peptic-related activity is responsible for gastric ulcer occurrence. The causes of hydrochloric/peptic aggression are multiple and varied: excessive acidity and altered mucous barrier, gastric mucosa hypoperfusion with hypoxia, mucosa metaplasia, gastritis, and marginal area (MALT) lymphomas. All these changes may, individually or cumulatively, lead to gastric ulcerous lesions. It is not our intention to analyse the etiopathogenesis of such pathology, but one thing is certain and unanimously accepted: the gastric mucosa becomes vulnerable to hydrochloric-peptic aggression.

The histology of ulcers was first described by Jean Cruveilhier, a student of Guillaume Dupuytren. In 1835, J. Cruveilhier provided a well-known description of gastric ulcer microscopy, and this is why, for a long time, ulcer lesions were known as Cruveilhier ulcers [1]. Over time, an entire series of factors generating ulcer lesions were defined, such as non-steroid anti-inflammatories (NSAIDs), stress ulcers, cortisone-induced ulcers, lymphoproliferative conditions, immunosuppressant therapy, chemotherapy, etc. The classic anatomy and pathology definition of an ulcer, however, comprises three microscopic components:Deterioration of the mucosa and the entire gastric wall, caused by necrosis, that can progress to the serous membrane, thus generating perforation of the stomach;An inflammatory lymphoplasmacytic process at the bottom of the ulcer and its margins;Microvascular changes in the ulcer’s perimeter with arteriole hyalinosis.

How do the structural changes occur in a strictly circumscribed area of the gastric wall? Chronic gastritis in different forms: *Helicobacter pylori* (HP) gastritis, biliary gastroduodenal reflux gastritis, lymphocytic gastritis, and autoimmune gastritis. For example, *Helicobacter pylori* are present in 60% of the healthy population, whereas gastric ulcers are present in 0.20% of these, i.e., 0.02% of the population [2,3,4,5]. Does this finding support the causal relation between *Helicobacter pylori* and the incidence of gastric ulcers? Researchers are now increasingly paying attention to the changes in gastric submucosa blood microcirculation [5,6,7,8,9]. Virchow and Aschoff were the pioneers that found that gastric wall haemodynamic alterations are responsible for the occurrence of ulcers. Their theory, revived nowadays, can provide satisfactory explanations of the shape and location of gastric ulcers [8]. This field of research showed that submucosa vascular plexuses, with a functional reduction in the gastric parietal blood flow in a limited section, result in a miniature focal infarction, generating ulcerating fibrinoid necrosis.

Based on these considerations, we discuss a series of unusual gastric ulcers: multiple ulcers and giant (over 2–3 cm) ulcers, some of them neoplastic, which manifest in significant acute bleeding or perforation of the wall. Multiple and giant gastric ulcers over 2–3 cm occur in 27% and 14% of cases, respectively [1]. Sometimes, surgical practice results in acute digestive bleeding that is impossible to stop endoscopically, which is a real surgical emergency. Gastric endoscopy may show the existence of single or multiple ulcer lesions. As the bleeding is severe and in full progression, with hemodynamic failure, endoscopic biopsy, endoscopic haemostasis, and TAE are contraindicated, the situation requiring a prompt surgical solution. We are often confronted, during surgery, with unusual ulcer lesions: ulcers over 2–3 cm, with a malignant ulcer appearance, multiple ulcers, etc. The surgical approach in the middle of a haemorrhage requires the removal of the bleeding lesion. Using in situ hemostasis is not indicated, as it poses the risk of some remaining malignant lesion or bleeding recurrence. In such situations, we are conditioned to use resection surgery. Because is not an ordinary gastric lesion, unique, and less than 2 cm, in situ haemostasis is not indicated. Such situations are encountered in ulcers of the gastric wall, greater curvature ulcers, multiple (more than two) ulcers, and giant ulcers (over 3–4 cm). In some situations, we must proceed to quasi-total gastric resections to remove the bleeding lesions. This approach can be used for multiple lesions. For giant ulcers (some over 5 cm), shaping gastric resections are required for preserving a receptor gastric segment. Next, we present the unusual types of gastric ulcers.

## 2. Giant Ulcers (over 2–3 cm)

These are ulcer lesions over 2–3 cm, often in atypical locations, on the stomach wall, or have greater curvature. Some of them are 5–6 cm, with bleeding and perforation complications obscured by a neighbouring organ (pancreas or liver) (Figure 1 and Figure 2). The appearance of these lesions sometimes suggests a neoplastic origin, and yet they are treated as benign. We have found lesions where radiology examination was suggestive of a gastric diverticulum or a gastric fundus tumoral process (Figure 3 and Figure 4). We have found such ulcers after immunosuppressant therapies with mycophenolate–mofetil administered post-heart transplant [10,11,12,13,14].

## 3. Multiple Ulcers

In general, there are three categories of ulcers that can be present based on the multiplicity of lesions: Cruveilhier chronic ulcers, acute ulcers, and lymphomatous ulcers. [9,10]. We had one patient with gastrointestinal haemorrhage, operated in an emergency, with over 100 typical diffuse ulcer lesions on the entire gastric surface, between 1–2 cm lesions and 3 cm ulcer craters, with the latter involving the fornix in a close subcardial position. A quasi-total gastrectomy was required for haemostasis (Figure 5). To our surprise, all of this patient’s ulcers, histologically, were Cruveilhier chronic ulcers. Another particular aspect, in this case, was the absence of gastric mucus, confirmed by PAS staining. Multiple ulcers can be detected after immunosuppressant therapies [10] and within the evolution of lymphocytic gastritis, characterized by the excessive lymphocytes infiltrate in the gastric mucosa and submucosa [15,16,17].

In the same category of multiple gastric ulcers, we found acute ulcers secondary to NSAID or aspirin treatment. These are usually present some areas of ulcerated mucous membranes, 3–30 mm in diameter, most often situated in the distal half of the stomach. These are plane, dish-like ulcers. They bleed easily and are not treatable by in situ haemostasis. Histology records the absence of inflammatory cells in the lesion crater and the ulcer’s vicinity. Gastritis-related changes are also often absent. Their dominant histological and pathological feature is necrosis [16,17,18].

Stress ulcers are often multiple in nature, in the form of multi-eroding gastritis, with clusters of ulcers or ulcers dispersed on a large portion of the gastric surface. Often, they do not extend beyond the mucosa. They occur under severe biological stress issues, with underlying organ failure, in various states of shock. Acute stress ulcers are difficult to control, severe, and accompanied by a significant mortality rate. Perforations are extremely rare. Such patients are usually hospitalized in an ICU. Such ulcers are under 1 cm in size, with supple (flexible and elastic), non-hardened margins, and without callosities. Additionally, the mucosa of the chorionic inflammatory reaction is missing. Prophylaxis is beneficial; thus, the incidence of haemorrhagic stress ulcers has currently decreased by 50%. However, the mortality rate of those with these ulcers is still high: 33% in ICU patients [19,20,21].

Multiple ulcers might also develop in the evolution of neoplastic conditions such as gastric lymphoma, having a separate share within ulcer pathology. Stomach damage may appear in combination with a large variety of gastrointestinal lymphomas, with or without lymph node involvement. Our area of interest only covers primitive gastric lymphomas, most of which are non-Hodgkin’s with a B lymphocyte substrate. However, ulcer lesions might also be found in association with MALT lymphomas. A lymphoma might be a primitive gastric lymphoma, manifesting as an ulcer lesion [19]. Lymphomatous ulcers, but thankfully do not involve ulcers over 2–3 cm and perforations or significant bleeding. Their depth is shallow, and they are superficially and frequently arranged in clusters in one gastric area (Figure 6, Figure 7 and Figure 8). Their macroscopic appearance is dish-like, with pseudo-diphtheroid white deposits. Ulcerating lymphomas surgery involves gastric resection with lymph node resection, adapted to lesion location, from D1 to D3/D4 [19]. This surgical technique obeys the principles of lymph node surgery [18,22,23]. Gastric ulcers are versatile and polymorphic that they might pose serious technical difficulties. It is often impossible for the surgeon to reach a diagnosis based on the extemporaneous histology and pathology investigation of a biopsy specimen. In the absence of this technique, the bleeding ulcer lesion must be removed without the decisive support of histology and pathology diagnosis.

The literature describes some sporadic case studies or presentations of atypical (multiple or over 2–3 cm) gastric ulcers. The interest in this field manifested quite early [24,25,26,27]. However, over the past 20 years, few studies on this subject have been published.

Surgery was and still is capable of successfully treating the so-called untreatable, resistant to treatment, or reoccurring after conservative treatment ulcers. Such ulcers include multiple and ulcers over 2–3 cm, which are rare and atypical. The interest in these unusual ulcers appeared because of their features: a relatively low response to conservative drug treatment, severe complications (perforations and haemorrhage) in over 10% of the cases, as well as the increased risk of a malignant lesion. These characteristics are shared between ulcers over 2–3 cm and multiple gastric ulcers [28,29,30]. According to these authors, published over 50 years ago, approximately 29% of ulcers over 2–3 cm are malignant. More recent studies determined, however, that the malignancy rate of ulcers over 2–3 cm or multiple ulcers is not different from that of usual gastric ulcers [31,32]. Therefore, in an uncomplicated ulcer over 2–3 cm, 12 weeks of conservative treatment can be used to achieve a healing rate of 88% [31]. The remaining 12% of cases are referred for surgery. However, in the case of ulcers with complications such as perforations or haemorrhage, conservative treatment is not possible.

It was determined that ulcer lesions are only the final expression of various nosologic entities. Ulcers associated with bleeding only complicate the therapeutic approach and the selection of the approach (endoscopy or classic surgery). The presence of the haemodynamic response of bleeding limits the therapeutic choices. The surgical approach is predominant in such situations. The interoperative discovery of multiple or ulcers over 2–3 cm, as well as of lesions with unusual locations, will lead to difficulties in choosing the tactics and technique indications. The choice between benign and malignant is most important. The objectives are haemostasis and lesion removal. After that, the surgical team is free to choose the approach and the extent of the surgery. Such unusual situations are not subject to therapy protocols that allow for a documented choice.

Twenty-five years ago, some of us were invited to a gastroenterology congress where the new treatments at eradicating *Helicobacter pylori* were debated. At the time, the failures of surgery, which mutilated the stomach and invalidated the pylorus, were also highlighted, which create the conditions for the appearance of intestinal gastric metaplasia, a precancerous condition. The criticism against the surgeons was obvious and undisguised.

The conclusion was that the anti-HP treatment cures 75–80% of gastro-duodenal ulcers. Surgeons deal with the 20–25% cases of ulcers where conservative treatment proved to be inefficient. A total of 5–10% of the surgical cases develop residual postoperative issues, i.e., operated stomach conditions. Surgical treatment is the last-resort therapy for cases where all other methods failed.

## 4. Conclusions

For unusual yet uncomplicated ulcers, whether ulcers over 2–3 cm or multiple ulcers, we think that the only secure approach remains gastric resection as the first choice. Simple endoscopic haemostasis or suture approaches involve, in these particular cases, accepting a complicated subsequent evolution. The first choice surgery thus avoids both the exceedingly more frequent complications of such ulcers and the potential malignancy of the lesion.

## Figures and Tables

**Figure 1 medicina-57-01345-f001:**
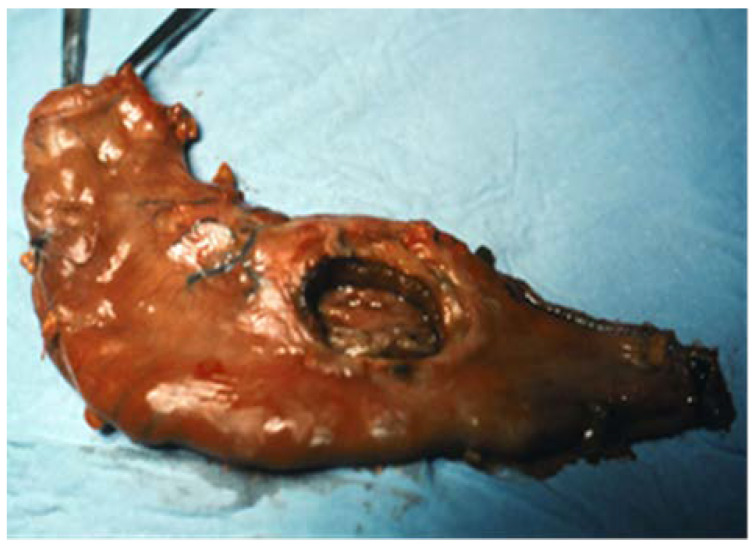
Quasi-total gastrectomy piece for a benign chronic giant ulcer (5 cm), in the median third of the stomach, penetrative towards the liver.

**Figure 2 medicina-57-01345-f002:**
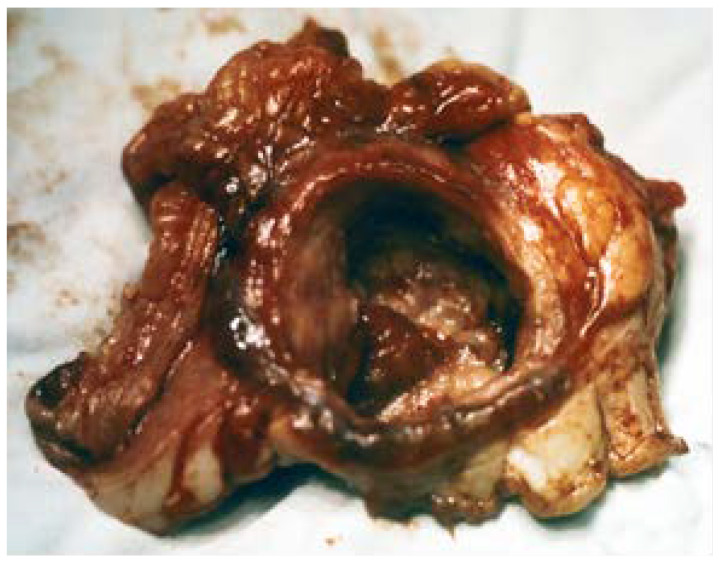
Benign giant ulcer, 4 cm in diameter, penetrative towards the pancreas.

**Figure 3 medicina-57-01345-f003:**
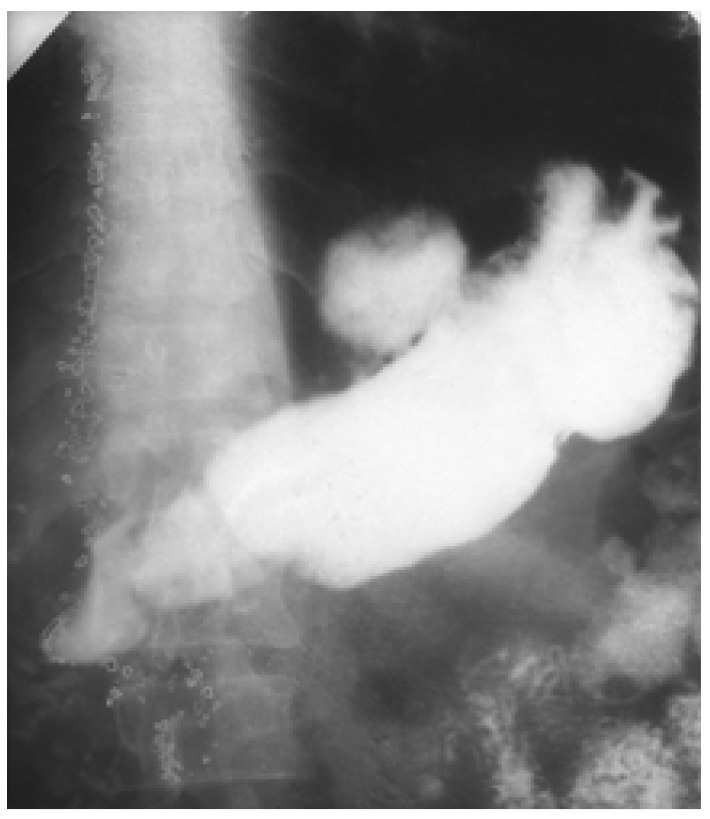
Barium meal. Giant ulcer in the median third of the stomach, with a pseudodiverticular subcardial lumen deformity.

**Figure 4 medicina-57-01345-f004:**
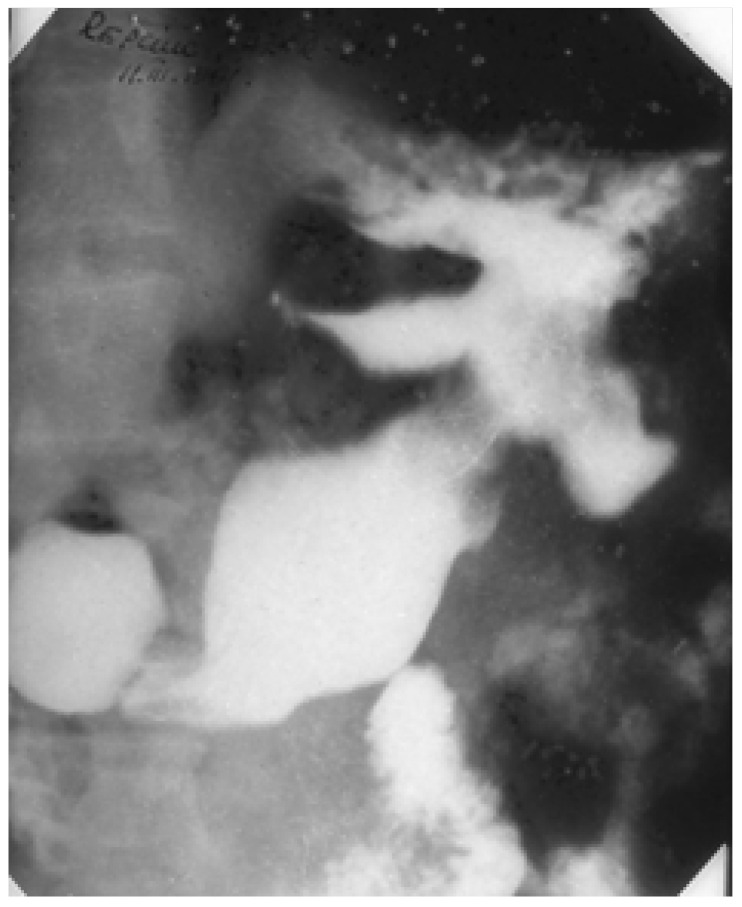
Barium meal. Giant ulcer with a clover-leaf deformity in the median third of the stomach, a pseudoneoplastic benign giant ulcer.

**Figure 5 medicina-57-01345-f005:**
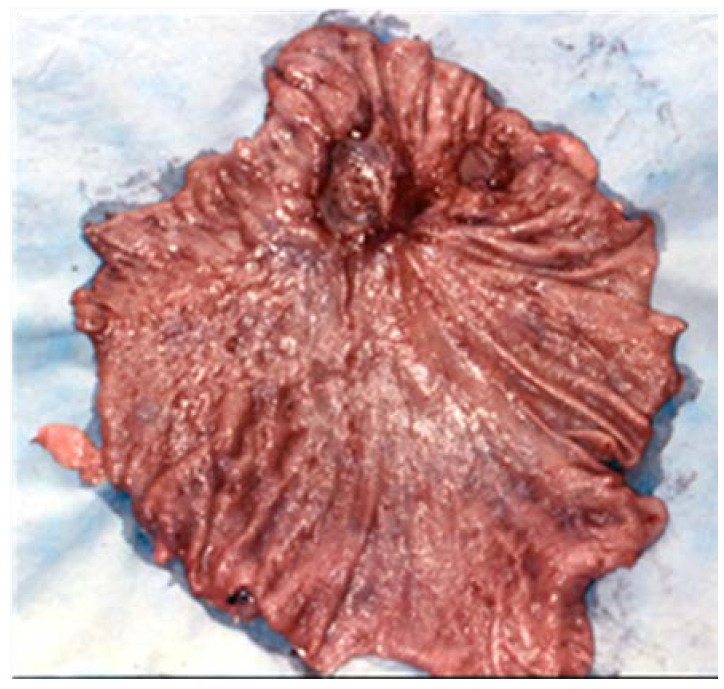
Quasi-total gastrectomy piece for multiple ulcers, diffuse over the gastric surface, 1–2 mm to 3 cm in diameter.

**Figure 6 medicina-57-01345-f006:**
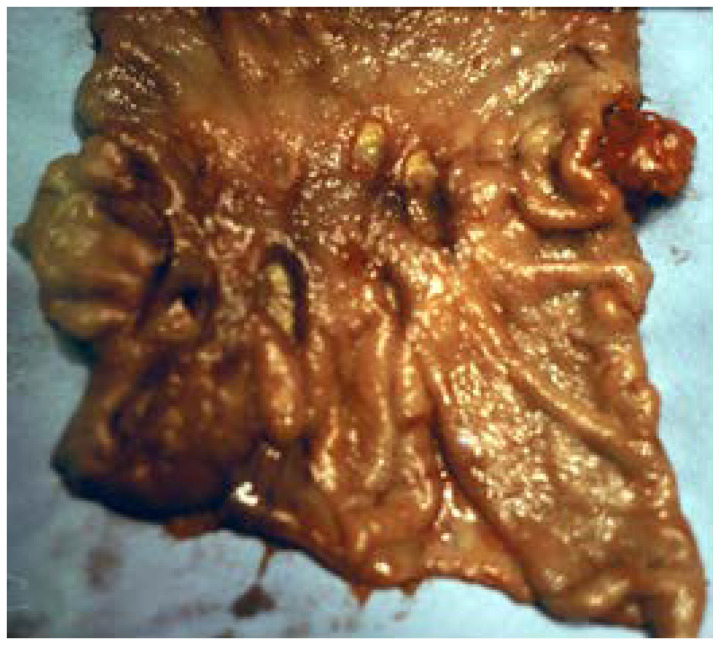
Ulcerating gastric lymphoma. Surgery piece: 2/3 gastric resection for multiple lymphomatous ulcers.

**Figure 7 medicina-57-01345-f007:**
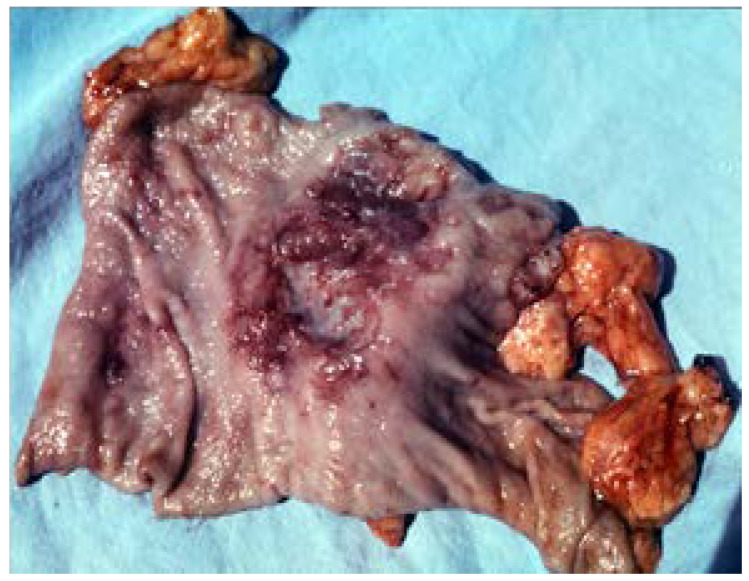
Bleeding giant ulcer occurring in the evolution of an ulcerating gastric lymphoma.

**Figure 8 medicina-57-01345-f008:**
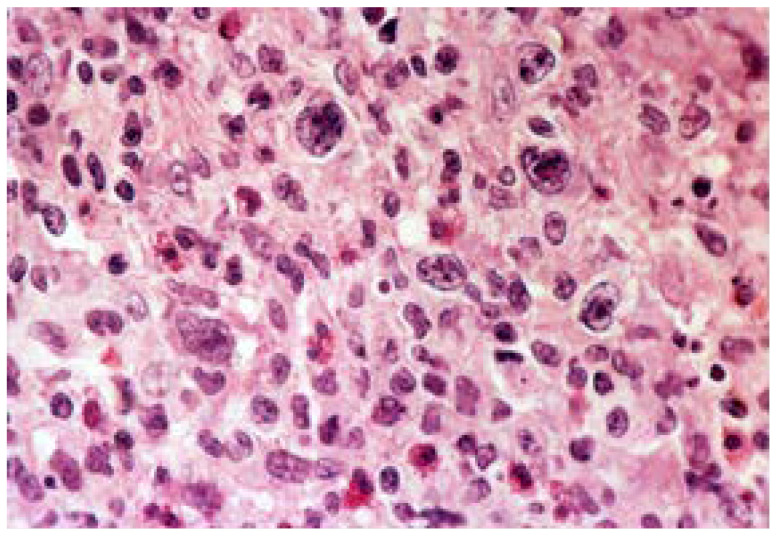
Histology imaging of a primitive, non-Hodgkin, large cell gastric lymphoma Van Gieson staining, ×40.

## Data Availability

Not applicable.

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
