# Peer review of "Unusual Complicated Gastric Ulcers"

_medicina, 2021, doi:10.3390/medicina57121345_

Round 1
Reviewer 1 Report
I congratulate the authors for their proper review work.Suggestions:
-I suggest modifying the expression "epiploonoplasty" by the expression "epiploplasty" (Intrduction section)
-Decimal figures must be expressed in the same way throughout the text. -Bibliographic references must be expressed using a hyphen (2-5), not commas (2,3,4,5).
-I suggest defining the giant ulcer in the same way throughout the introduction (example: more than 2-3 cm).
-I suggest replacing the expression "Barium meal" with the expression "Barium swallow study" in the feet of figures three and four.
-I suggest replacing the expression "... callosities that do not exist" by the expression "... without callosities" (section 3. Multiple ulcers).
-I suggest replacing the expression "thankfully" by the expression "fortunately" (section 3. Multiple ulcers).
-I suggest replacing the expression "According to these authors..." by the expression "According to some authors..." (section 3. Multiple ulcers).
-I think there is an error quoting reference 32: "... the usual gastric ulcers (31, 232) (section 3. Multiple ulcers).
-References must be expressed according to the journal's standards (https://www.mdpi.com/journal/medicina/instructions).
Author Response
Response to Reviewer 1 Comments
Point 1:
Please see the attachment
Response 1:
Please provide your response for Point 1. (in red)
Open Review
(x) I would not like to sign my review report
( ) I would like to sign my review report
English language and style
( ) Extensive editing of English language and style required
( ) Moderate English changes required
(x) English language and style are fine/minor spell check required
( ) I don't feel qualified to judge about the English language and style
Is the work a significant contribution to the field?
Is the work well organized and comprehensively described?
Is the work scientifically sound and not misleading?
Are there appropriate and adequate references to related and previous work?
Is the English used correct and readable?
Comments and Suggestions for Authors
I congratulate the authors for their proper review work.
Suggestions:
-I suggest modifying the expression "epiploonoplasty" by the expression "epiploplasty" (Intrduction section)
-Decimal figures must be expressed in the same way throughout the text. -Bibliographic references must be expressed using a hyphen (2-5), not commas (2,3,4,5).
-I suggest defining the giant ulcer in the same way throughout the introduction (example: more than 2-3 cm).
-I suggest replacing the expression "Barium meal" with the expression "Barium swallow study" in the feet of figures three and four.
-I suggest replacing the expression "... callosities that do not exist" by the expression "... without callosities" (section 3. Multiple ulcers).
-I suggest replacing the expression "thankfully" by the expression "fortunately" (section 3. Multiple ulcers).
-I suggest replacing the expression "According to these authors..." by the expression "According to some authors..." (section 3. Multiple ulcers).
-I think there is an error quoting reference 32: "... the usual gastric ulcers (31, 232) (section 3. Multiple ulcers).
-References must be expressed according to the journal's standards (https://www.mdpi.com/journal/medicina/instructions).
Submission Date
20 October 2021
Date of this review
31 Oct 2021 11:18:14
© 1996-2021 MDPI (Basel, Switzerland) unless otherwise stated
Thank you for the tips and guidance useful in making the article
Response to Reviewer 1 Comments
- I suggest modifying the expression "epiploonoplasty" by the expression "epiploplasty" (Intrduction section)
- I changed the word; „Epiploonoplasty" by the expression "epiploplasty" (Introduction section)
- Decimal figures must be expressed in the same way throughout the text. -Bibliographic references must be expressed using a hyphen (2-5), not commas (2,3,4,5).
- - I corrected Decimal figures
- I suggest defining the giant ulcer in the same way throughout the introduction (example: more than 2-3 cm).
- I replaced and introduced the same specification "giant ulcers" with the suggested wording more than 2-3cm
- I suggest replacing the expression "Barium meal" with the expression "Barium swallow study" in the feet of figures three and four.
- I replaced the expression "Barium meal" with the expression "Barium swallow study" in the feet of figures three and four.
- I suggest replacing the expression "... callosities that do not exist" by the expression "... without callosities" (section 3. Multiple ulcers).
- I replaced the expression "... callosities that do not exist" by the expression "... without callosities" (section 3. Multiple ulcers).
- I suggest replacing the expression "thankfully" by the expression "fortunately" (section 3. Multiple ulcers).
- I replaced the expression "thankfully" by the expression "fortunately" (section 3. Multiple ulcers).
- I suggest replacing the expression "According to these authors..." by the expression "According to some authors..." (section 3. Multiple ulcers).
- I replaced the expression "According to these authors..." by the expression "According to some authors..." (section 3. Multiple ulcers).
- I think there is an error quoting reference 32: "... the usual gastric ulcers (31, 232) (section 3. Multiple ulcers).
- I changed the usual gastric ulcers (31, 232) (section 3. Multiple ulcers) with (31,32)
- References must be expressed according to the journal's standards (https://www.mdpi.com/journal/medicina/instructions).
- I corrected References according to the journal's standards (https://www.mdpi.com/journal/medicina/instructions).
- I sent the article to the magazine for translation „ [English ID: english-37189] English pre-editing”

Reviewer 2 Report
In this review, the authors aimed at drawing attention to a practical issue, i.e., the approach for certain 10 unusual gastric ulcers, with haemorrhage- or perforation-induced complications. The authors discussed the circumstances determining the occurrence of such ulcer lesions, their diverse aetiology and pathogenesis, their common manifestations, and the severity of their evolution.
The written English is not so good. Please ask professional editing service for English writing.
The authors classified unusual ulcers as 2 categories: giant and multiple. It is difficult to understand why these are the types of unusual ulcer categories.
About giant ulcers, the authors described that they have found such ulcers after immunosuppressant therapies with Mycophenolate-Mofetil administered post-heart transplant. In these cases, PPI or H2RA are not used for preventing ulcer formation?
For malignant lymphoma of the stomach, surgery is not the first treatment choice. Usually giant ulcers caused of malignant lymphoma are cured by the anti-lymphoma therapy such as CHOP therapy. How is the treatment strategy for these ulcers constructed in your country?
The resolution of Figure 5 is low. So I cannot see the multiple ulcers well. Would you mark on the picture for indicating multiple ulcers?
Author Response
Response to Reviewer 2 Comments
Point 1:
Response 1:
Please provide your response for Point 1. (in red)
Open Review
(x) I would not like to sign my review report
( ) I would like to sign my review report
English language and style
( ) Extensive editing of English language and style required
(x) Moderate English changes required
( ) English language and style are fine/minor spell check required
( ) I don't feel qualified to judge about the English language and style
Is the work a significant contribution to the field?
Is the work well organized and comprehensively described?
Is the work scientifically sound and not misleading?
Are there appropriate and adequate references to related and previous work?
Is the English used correct and readable?
Comments and Suggestions for Authors
In this review, the authors aimed at drawing attention to a practical issue, i.e., the approach for certain 10 unusual gastric ulcers, with haemorrhage- or perforation-induced complications. The authors discussed the circumstances determining the occurrence of such ulcer lesions, their diverse aetiology and pathogenesis, their common manifestations, and the severity of their evolution.
The written English is not so good. Please ask professional editing service for English writing.
The authors classified unusual ulcers as 2 categories: giant and multiple. It is difficult to understand why these are the types of unusual ulcer categories.
About giant ulcers, the authors described that they have found such ulcers after immunosuppressant therapies with Mycophenolate-Mofetil administered post-heart transplant. In these cases, PPI or H2RA are not used for preventing ulcer formation?
For malignant lymphoma of the stomach, surgery is not the first treatment choice. Usually giant ulcers caused of malignant lymphoma are cured by the anti-lymphoma therapy such as CHOP therapy. How is the treatment strategy for these ulcers constructed in your country?
The resolution of Figure 5 is low. So I cannot see the multiple ulcers well. Would you mark on the picture for indicating multiple ulcers?
Submission Date
20 October 2021
Date of this review
18 Nov 2021 15:06:02
© 1996-2021 MDPI (Basel, Switzerland) unless otherwise stated
Thank you for the tips and guidance useful in making the article
Response to Reviewer 2 Comments
Point 1:
Response
- In this review, the authors aimed at drawing attention to a practical issue, i.e., the approach for certain 10 unusual gastric ulcers, with haemorrhage- or perforation-induced complications. The authors discussed the circumstances determining the occurrence of such ulcer lesions, their diverse aetiology and pathogenesis, their common manifestations, and the severity of their evolution.
- In the article we presented only surgical emergencies with severe hemorrhage or perforation
- The written English is not so good. Please ask professional editing service for English writing.
- I sent the article to the magazine for translation „ [English ID: english-37189] English pre-editing”
- The authors classified unusual ulcers as 2 categories: giant and multiple. It is difficult to understand why these are the types of unusual ulcer categories.
- Giant ulcers (over 2cm) and multiple ones are "unusual" because they are „particular”, due to their rarity. About 7% of ulcers are giant (≥ 2cm). Giant ulcers are also unusual in that they have a predominantly complicated course of hemorrhage and / or perforation with peritonitis (Akira M., J A Eaden - Gastrointestinal: giant gastric ulcers J Gastroenterol Hepatol. 2001 May;16(5):573. doi: 10.1046/j.1440-1746.2001.02488.x.). Also, the failure rate of conservative treatment is high, requiring a surgical approach.
- Multiple throat ulcers are found in a percentage of 2-4% according to statistics provided by Dolphin J.A. The multiplicity of lesions gives them the unusual character.
- About giant ulcers, the authors described that they have found such ulcers after immunosuppressant therapies with Mycophenolate-Mofetil administered post-heart transplant. In these cases, PPI or H2RA are not used for preventing ulcer formation?
- There is no recommendation that the administration of Mycophenolate-Mofetil be associated with gastric protection by PPI or H2RA. The side effects of MOFETIL are extremely low in terms of gastric damage. Mofetil is given as gastro-resistant tablets with slow intestinal resorption and is insoluble in the stomach.
- For malignant lymphoma of the stomach, surgery is not the first treatment choice. Usually giant ulcers caused of malignant lymphoma are cured by the anti-lymphoma therapy such as CHOP therapy. How is the treatment strategy for these ulcers constructed in your country?
- The approach is similar in Romania, the treatment of gastric lymphomas is not primarily surgery. But, as we mentioned in the introduction, our group of patients included only giant or multiple ulcers complicated by severe hemorrhage or perforation, real urgent indications for surgery as the first choice of treatment, in the absence of endoscopic biopsies.
- The resolution of Figure 5 is low. So I cannot see the multiple ulcers well. Would you mark on the picture for indicating multiple ulcers?
- In figure 5 on each fold of the gastric mucosa were ulcers with dimensions larger than 1mm. The image is not favourable to support the multitude of ulcerous lesions with dimensions from 1-2mm to over 3cm. The quality of the photographic image cannot sufficiently document our statement. I mention that there were hundreds of ulcerative lesions lined up on each gastric fold. Trying to enlarge the image distorts it. I'm going to draw some arrows to indicate some ulcers.

Round 2
Reviewer 2 Report
Thank you for your response and English editing.